# Concerted Regulation of Glycosylation Factors Sustains Tissue Identity and Function

**DOI:** 10.3390/biomedicines10081805

**Published:** 2022-07-27

**Authors:** Daniel Sobral, Rita Francisco, Laura Duro, Paula Alexandra Videira, Ana Rita Grosso

**Affiliations:** 1Associate Laboratory i4HB—Institute for Health and Bioeconomy, NOVA School of Science and Technology, Universidade NOVA de Lisboa, 2829-516 Caparica, Portugal; rab.francisco@campus.fct.unl.pt (R.F.); l.duro@campus.fct.unl.pt (L.D.); 2UCIBIO—Applied Molecular Biosciences Unit, Department of Life Sciences, NOVA School of Science and Technology, Universidade NOVA de Lisboa, 2829-516 Caparica, Portugal; 3CDG & Allies—Professionals and Patient Associations International Network (CDG & Allies—PPAIN), Life Sciences Department, NOVA School of Science and Technology (FCT NOVA), Universidade NOVA de Lisboa, 2829-516 Caparica, Portugal

**Keywords:** glycosylation machinery, genomics, transcriptomics, healthy tissues, cancer

## Abstract

Glycosylation is a fundamental cellular process affecting human development and health. Complex machinery establishes the glycan structures whose heterogeneity provides greater structural diversity than other post-translational modifications. Although known to present spatial and temporal diversity, the evolution of glycosylation and its role at the tissue-specific level is poorly understood. In this study, we combined genome and transcriptome profiles of healthy and diseased tissues to uncover novel insights into the complex role of glycosylation in humans. We constructed a catalogue of human glycosylation factors, including transferases, hydrolases and other genes directly involved in glycosylation. These were categorized as involved in N-, O- and lipid-linked glycosylation, glypiation, and glycosaminoglycan synthesis. Our data showed that these glycosylation factors constitute an ancient family of genes, where evolutionary constraints suppressed large gene duplications, except for genes involved in O-linked and lipid glycosylation. The transcriptome profiles of 30 healthy human tissues revealed tissue-specific expression patterns preserved across mammals. In addition, clusters of tightly co-expressed genes suggest a glycosylation code underlying tissue identity. Interestingly, several glycosylation factors showed tissue-specific profiles varying with age, suggesting a role in ageing-related disorders. In cancer, our analysis revealed that glycosylation factors are highly perturbed, at the genome and transcriptome levels, with a strong predominance of copy number alterations. Moreover, glycosylation factor dysregulation was associated with distinct cellular compositions of the tumor microenvironment, reinforcing the impact of glycosylation in modulating the immune system. Overall, this work provides genome-wide evidence that the glycosylation machinery is tightly regulated in healthy tissues and impaired in ageing and tumorigenesis, unveiling novel potential roles as prognostic biomarkers or therapeutic targets.

## 1. Introduction

Glycosylation is a complex multi-enzymatic process that includes assembling monosaccharides into glycans and transferring these onto proteins or lipids [1], with recent evidence suggesting small RNAs can also be glycosylated [2]. Most human proteins are thought to be glycosylated [3], influencing all aspects of cellular activity, including immunity [4,5], cell signaling [6], and cell adhesion [7]. Glycosylation is a ubiquitous post-translational modification present in all domains of life [8], enabling the generation of a large diversity of molecular structures, not directly bound by genetic information, much like the widely studied phosphorylation.

The biosynthesis of distinct glycoconjugates can be classified in different glycosylation pathways depending on the site of the glycosidic linkage and the glycan attached, namely: N-linked glycosylation, O-linked glycosylation; Glycosaminoglycans (GAG); Glypiation (GPI); and lipid glycosylation [9]. In humans, approximately hundreds of glycosylation factors are responsible for this post-translational modification process, including specific transferases (acetylglucosaminyltransferase, sialyltransferase, mannosyltransferase, N-acetylgalactosaminyltransferase, galactosyltransferase, fucosyltransferase, glucosyltransferase, xylosyltransferase), hydrolases (mannosidase, sialidase, glucosidase, fucosidase) and enzymes involved in monosaccharide precursor synthesis and transport [1]. Each glycosylation pathway is accomplished by a specific and distinct set of these enzymes, where some participate in different glycosylation pathways. A precise and tightly regulated action of the different enzymes and proteins are demanded to produce the glycoconjugate profiles specific and distinctive of each cell type and tissue and each development/functional stage [10,11].

Unsurprisingly, dysfunction of glycosylation factors leads to many pathologies [12], including the rare set of congenital disorders of glycosylation (CDG), cancer, metabolic and inflammatory diseases [12,13,14]. Notably, since aberrant glycosylation plays an important role in the immunosuppression of tumors and in the setting up of inflammatory diseases, they are used in the design of novel immunotherapies [15] or as biomarkers [16]. The most common perturbations in cancer are increased sialylation or fucosylation, O-glycan truncation, and N- and O-linked glycan branching [17]. Increased sialylation, in epithelial cancers, promotes immune evasion [18] and contributes to tumor progression and poor prognosis [19].

Our understanding the role of aberrant glycosylation in dysregulating organismal and cellular homeostasis is significantly progressing. Still, many steps remain largely uncharacterized, hindered by the inherent complexities of glycosylation. Large consortia have been generating a wide diversity of molecular profiles, in both healthy (e.g., GTEX [20], Protein Atlas [21]) and diseased tissues (e.g., TCGA [22]). However, these data have not yet been comprehensively examined to elucidate the role of glycosylation factors. In this study, we aimed at exploring available human molecular profiles, integrating them to obtain novel insights into the complex role of the glycosylation machinery. Our genome/transcriptome integrative analysis of a catalogue of 242 human glycosylation factors revealed that this ancient family of genes have expanded under some evolutionary constraints, capacitating glycosylation factors in defining tissue identity through a tissue-specific glycosylation code. Moreover, our work suggests that glycosylation-driven tumorigenesis is mediated mostly by large genomic amplifications and corresponding transcriptome alterations in glycosylation factors. Overall, our findings reveal novel unappreciated potential roles of glycosylation in human health.

## 2. Materials and Methods

### 2.1. Selection of Glycosylation-Factors and Gene Expression-Related Genes

A list of 242 glycosylation factors (Appendix A) was obtained by collecting information from: GGDB glycosylation factor Database [23]; Reactome [24] (N-linked glycosylation; O-linked glycosylation); KEGG [25] (N-Glycan biosynthesis; Mucin and Mannose type O-glycan biosynthesis); and selecting human genes annotated in ENSEMBL [26] release 104 (May 2021) with the Gene Ontology term “protein glycosylation” (GO:0006486). The list was then manually curated to correct discontinued official gene symbols and remove genes encoding proteins that may undergo glycosylation but with no evidence for a direct role in the glycosylation process itself.

The genes encoding for regulators of gene expression were retrieved from different sources. Briefly, a list of 287 epigenetics factors was extracted from Boukas et al. [27]. A list of 653 transcription factors was extracted from Jolma et al. [28], including only those genes with a known motif, and excluding the ones only containing C2H2 zinc fingers. Splicing (326 genes), translation (154 genes), phosphorylation (672 genes) and ubiquitination factors (619 genes) were selected based on the GO annotation in ENSEMBL (GO:0008380, GO:0006417, GO:0006468 and GO:0016567, respectively).

### 2.2. Phylogenetic Information and Expression

Ortholog and paralog information was downloaded from ENSEMBL Biomart. Information on the paralogous first duplication event was compressed to chordates (everything before Vertebrata), vertebrates (everything before Mammalia), and mammals (everything else except humans). To analyze phylogenetic conservation of tissue expression, multi-species expression data were obtained from Brawand et al. [29], comparing six organs across ten species. Namely, we used the normalized RPKM from constitutive exons of 1-to-1 orthologs. For each gene, we scaled the expression and obtained the standard deviation of the tissue with the minimum standard deviation between species.

### 2.3. Transcriptome Profiles of Healthy Tissues

The expression levels for healthy tissues were obtained from Protein Atlas and Genotype-Tissue Expression (GTEx) Project Protein atlas gene expression data were downloaded from https://www.proteinatlas.org/download/rna_tissue_hpa.tsv.zip (downloaded 28 October 2019), and the normalized NX values were used for analyses. The GTEX data comprised transcriptome profiles of 28 healthy tissues and were obtained from the GTEx Portal on 28 October 2019. Only public metadata on age and sex were used, where age was classified in 10-year blocks. Moreover, transformed cells were excluded from the analysis. The normalized TPM values of GTEX samples were used for downstream analyses. We used the UMAP R package with GTEX expression data for UMAP analyses.

### 2.4. Tissue-Specificity and Classification Based on Transcriptome Profiles

To estimate the tissue predictive ability of glycosylation factors and remaining gene sets we used random forest models as implemented in the randomForest R package [30]. Briefly, samples of each tissue were divided into 75% train and 25% test. Then, 100 random forest models were used, each using the GTEX normalized expression data (TPMs) for 30 randomly selected genes (from the specific group of genes of interest). Finally, the Matthews Correlation Coefficient (MCC) was calculated based on the results of the classification of the test data.

To estimate tissue specificity, we used the TissueEnrich R package [31], with protein atlas data and also with GTEX data (Appendix A). In brief, a Tissue Enriched gene has at least five-fold higher expression levels in a particular tissue compared to all other tissues; a Group Enriched gene has at least five-fold higher expression levels in a group of 2–7 tissues compared to all other tissues; a Tissue Enhanced gene does not fall in the previous categories but has at least five-fold higher expression levels in a particular tissue compared to the average expression of all other tissues.

To estimate age and sex-specific expression, for each tissue and gene, we fitted a linear model of the GTEX normalized TPM gene expression of a given tissue depending on age (in decades) and sex (except for tissues of the reproductive system with samples of only one specific sex, where only age was used). The *p*-values derived from each gene’s linear model were corrected for multiple testing using the Benjamini–Hochberg procedure. Genes were age or sex-dependent if their coefficient in the linear model had an adjusted *p*-value less than 0.05.

### 2.5. Molecular Profiles of Tumor Samples

We downloaded public TCGA data from the GDC portal for all the 33 cancer types: ACC (adrenocortical carcinoma); BLCA (bladder urothelial carcinoma); BRCA (breast invasive carcinoma); CESC (cervical squamous cell carcinoma and endocervical adenocarcinoma); CHOL (cholangiocarcinoma); COAD (colon adenocarcinoma); DLBC (Lymphoid Neoplasm Diffuse Large B-cell Lymphoma); ESCA (Esophageal carcinoma); GBM (Glioblastoma multiforme); HNSC (head and neck squamous cell carcinoma); KICH (kidney Chromophobe); KIRC (kidney renal clear cell carcinoma), KIRP (kidney renal papillary cell carcinoma); LAML (Acute Myeloid Leukemia); LGG (Brain Lower-Grade Glioma); LIHC (liver hepatocellular carcinoma); LUAD (lung adenocarcinoma); LUSC (lung squamous cell carcinoma), MESO (Mesothelioma); OV (Ovarian serous cystadenocarcinoma); PAAD (pancreatic adenocarcinoma); PCPG (pheochromocytoma and Paraganglioma); PRAD (prostate adenocarcinoma); READ (rectum adenocarcinoma); SARC (sarcoma); SKCM (Skin Cutaneous Melanoma); STAD (Stomach adenocarcinoma); TGCT (Testicular Germ Cell Tumors); THCA (thyroid carcinoma); THYM (Thymoma); UCEC (uterine Corpus Endometrial Carcinoma); UCS (Uterine Carcinosarcoma); and UVM (Uveal Melanoma). The data included: clinical metadata, namely the patient and follow up tables; somatic mutations as MuTect2 Variant Aggregation and Masking maf files; copy number as GISTIC copy number focal scores text files; and RNA-Seq read counts as HTSeq count tables.

### 2.6. Impact of Genetic Alterations

To assess the genetic alterations associated with pathogenic phenotypes we used the Clinvar variant summary data downloaded from https://ftp.ncbi.nlm.nih.gov/pub/clinvar/ (release of 2 December 2019). Information on disease description was used with the wordcloud R package to produce a word cloud representing the relative frequency of terms in the Clinvar entries associated with glycosylation factors. The intolerance to loss of function mutations was obtained through the pLI score, that derives from the probability that a loss of function mutation occurs in a large cohort of healthy persons. The pLI scores were downloaded from Exact (ftp://ftp.broadinstitute.org/pub/ExAC_release/release0.3/functional_gene_constraint/fordist_cleaned_exac_r03_march16_z_pli_rec_null_data.txt), accessed on 4 April 2019. We also downloaded the list of cancer drivers from IntOgen release 1 February 2020. To identify genes under selective pressure in cancer, we applied the dNdScv method [32] using somatic mutation data from each of the TCGA cancer types. Briefly, dNdScv compares, for each gene, the observed tumor-specific mutations against a global background of mutations). A gene was considered under selection if it had significant evidence with dNdScv in at least one cancer type.

Influence of genomic perturbations in cancer patient survival was analyzed by Kaplan–Meier curve comparison using a log-rank test and a multivariate Cox proportional hazards analysis, as implemented in the survival R package [33].

### 2.7. Transcriptome Alterations in Tumor Samples

We used the HTSeq count tables to obtain normalized CPM values through TMM normalization in edgeR [34]. To infer gene expression perturbation in tumors, we used expression data of TCGA tumor samples and their paired normal samples (in cancers where both were available, namely BLCA, BRCA, CHOL, COAD, ESCA, HNSC, KICH, KIRC, KIRP, LIHC, LUAD, LUSC, PRAD; READ, STAD, THCA and UCEC) to perform a per-cancer differential expression analysis with the limma-voom R package [35] Genes with adjusted *p*-value lower than 0.05 were considered differentially expressed. Gene set enrichment analysis of gene expression pathways was performed using the fgsea package [36].

### 2.8. Association with Cellular Composition in Healthy and Diseased Tissues

To obtain estimates of the relationship between glycosylation and immune cell-populations in TCGA tumor samples, we downloaded the pre-calculated frequencies from TCIA (quantiseq frequencies) [37]. We evaluated the associations using Pearson’s correlation and lasso regression. First, for each cancer, and glycosylation factor, we estimated the Pearson correlation between the normalized gene expression and immune cell population frequencies. We then counted—for each gene—with how many tumor types each immune cell type correlated significantly (*p* < 0.05). We also applied the same method with normal GTEX tissues. For this, we inferred the frequencies of immune cell populations of GTEX samples by applying quantiseq [38] to all GTEX samples using the normalized TPM expression values. Lasso regression analysis was performed as described previously [39]. Briefly, the estimated frequency of M2 macrophages for each individual cancer sample (grouped by cancer type) was modelled by Lasso regression as implemented in the glmnet R package [40], with a 10-fold cross validation to choose the lambda parameter. Statistical significance of the explained variance by each model was assessed for values greater than zero using a margin of more than one standard deviation. Finally, the obtained models were evaluated by assessing the correlation (Pearson method) between the observed and predicted M2 levels based on tumor mutation and expression profiles. A similar approach based on Lasso Models was applied to detect alterations in glycosylation factors associated with the number of tumor subclones, i.e., intra-tumor heterogeneity. Intra-tumoral heterogeneity (ITH) of TCGA samples was defined as the number of clones estimated using EXPANDS [41] on the tumor SNVs and CNAs.

### 2.9. Statistical Analysis

All statistical analysis and figures were generated using R. Besides the already mentioned R packages, we also used the graph package for the network plots; the ComplexHeatmap package to produce the oncoprint; the corrplot package heatmap with correlations; and the pheatmap package for the remaining heatmaps.

## 3. Results

### 3.1. Glycosylation Factors Are an Evolutionarily Conserved Family of Genes

We constructed a catalogue of 242 human genes directly involved in the process of glycosylation (see details in methods), herein referred to as glycosylation factors (Appendix A). Of these, 70 genes (29%) are mainly involved in N-linked glycosylation, 74 (31%) in O-linked glycosylation, 21 (9%) in glypiation (GPI), 30 (12 %) in lipid glycosylation, and 15 (6%) in glycosaminoglycan (GAG) glycosylation. The remaining 32 (13%) genes are involved in at least two of the above-mentioned pathways (Figure 1A). Regarding their functions, the vast majority (74%) of the glycosylation factors are glycosyltransferases (Appendix A), 10% are glycosyl hydrolases, with the remaining genes playing other roles (Appendix A).

**Figure 1 biomedicines-10-01805-f001:**
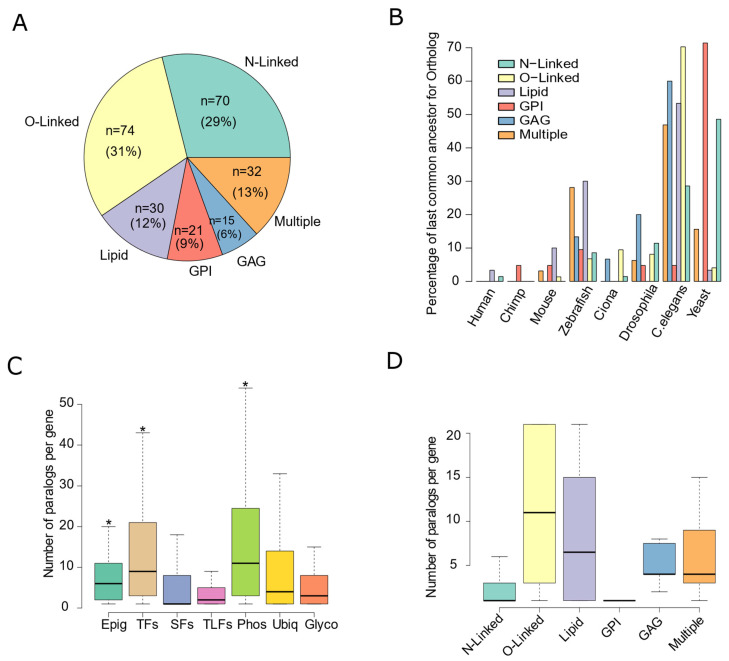
Glycosylation Factors are an evolutionarily conserved family of genes. (**A**) Pie chart with the frequencies of each glycosylation pathway among the 242 genes considered in this study: N-linked; O-Linked; Lipid; Glypiation (GPI); Glycosaminoglycans (GAG); Genes involved in more than one pathway (Multiple). (**B**) Barplot with the percentages of the organism representing the last common ancestor to the human gene, for each glycosylation subclass. (**C**) Boxplot of the total number of paralogous copies per gene, for each class of gene expression regulators: epigenetic (Epig); transcription (TF); Splicing (SF); Translation (TLF); Phosphorylation (Phos); Ubiquitination (Ubiq); Glycosylation (Glyco). An asterisk (*) represents that the number of paralogs per gene in the class is significantly different from glycosylation (Wilcoxon Test *p* < 0.05). (**D**) Boxplot of the total number of paralogous copies per gene for each glycosylation subclass.

Our phylogenetic analysis revealed that glycosylation is ubiquitous in the tree of life, with both N- and O-Glycosylation already appearing in archaeal and bacterial species [42,43]. In addition, most human glycosylation factors contain orthologs down to distant organisms such as the nematode worm (Figure 1B and Appendix A). In fact, some glycosylation subclasses emerged earlier, with 70% of the human glypiation genes and 50% of the N-linked glycosylation factors presenting orthologs already in yeast. Despite their early ancestry, glycosylation factors suffered several gene duplications increasing the number of paralogs present in the human genome (Appendix A). However, glycosylation factors have less paralogs than transcription, phosphorylation, and epigenetic factors (Figure 1C). In fact, 59% of N-linked glycosylation and 90% of glypiation genes are single copy, suggesting a strong control of gene copy number since their ancient origin (Figure 1D). Interestingly, O-linked and lipid glycosylation seem to be an exception, with the average number of paralogs per gene higher than 10 in the human genome (Figure 1D). A striking example is the GALNT gene family, required to initiate O-glycosylation, enclosing 21 human paralogs (Appendix A). Thus, glycosylation is a conserved and essential cellular process where most direct regulators have emerged early in evolution and were spared from large duplication events.

### 3.2. Glycosylation Factors Show Tissue-Specific Expression Preserved across Mammals

Given the complexity of the glycosylation machinery and the existence of cell-specific glycoconjugate profiles, we decided to explore how glycosylation factors expression varied across healthy tissues from the GTEX project [20]. An unsupervised analysis unveiled that human samples could be grouped according to tissue type based solely on the expression levels of glycosylation factors (Figure 2A), reinforcing the fact that each tissue may display different patterns of glycosylation. Hence, transcription profiles of glycosylation factors can distinguish the different tissues significantly better than most regulators of gene expression, except transcription and phosphorylation factors (Figure 2B). In fact, 25% of glycosylation factors present tissue-specificity, enclosing enriched or enhanced expression associated with a single tissue or a small group of tissues (Figure 2C and Appendix A). Such specificity is only surpassed by transcription and phosphorylation factor tissue-specific expression (Figure 2C). Interestingly, specificity appears to be most relevant in O-linked and lipid glycosylation pathways, where approximately 40% of the genes showed restricted expression to single or a small group of tissues (Appendix A). Concordantly, these gene subclasses also have the highest ability to distinguish different tissues (Appendix A). Thus, such findings support the existence of a tissue-specific glycosylation code that is sustained by activation of distinct glycosylation factors.

Within the different human tissues, the brain and digestive system presented the most distinct expression, with the highest number of tissue-specific glycosylation factors (Figure 2D and Appendix A). Relevant examples include *B4GAT1* and *B4GALNT1* transferases, showing brain-specificity (Figure 2D and Appendix A), in agreement with the fact that its disruption is known to cause defects in the nervous system [44,45]. Interestingly, the expression of paralogous glycosylation factors alternates and contributes to tissue-specific glycosylation. While *B4GALNT1* and *B4GALNT4* are brain specific, *B4GALNT2* and *B4GALNT3* are specifically expressed in tissues of the digestive system (Figure 2D). Or the case of *GALNT13* and *GALNT14* specifically expressed in the brain and kidney, respectively. Moreover, our analysis also identified signatures that may define distinct glycosylation patterns in tissues, highlighting: the brain (*GALNT9*, *GALNT13*, *ST8SIA3*, *B3GAT1*, *B3GAT2*, *MGAT5B*, *B4GAT1*, *B4GALNT1*, *COLGALT2*, *B4GALNT4* and *GALNT17*); stomach (*GALNT6*, *FUT9*, *A4GNT*, *B4GALNT3*); intestine (*ABO*, *B4GALNT2*, *GALNT8*, *ST6GALNAC1*); thyroid and parathyroid glands (*ST6GAL2*, *MGAT4C*, *GALNT18*, *ST6GALNAC3*, *GCNT1*) (Figure 2D, Appendix A).

To reinforce the existence of a tissue-specific code, we investigated how conserved is the expression of glycosylation factors across mammals. Interestingly, glycosylation factors unveiled a low within-tissue expression divergence of orthologs across different mammalian species, only slightly above transcription or phosphorylation factors (Appendix A). Moreover, the glycosylation subclasses with the highest tissue-specific expression, O-linked and lipid glycosylation, also revealed lower within-tissue expression variation across species (Appendix A). Such evolutionarily conserved tissue-specific profiles clearly strengthen the importance of glycosylation to maintain cell identity and tissue homeostasis in complex organisms. Thus, our results indicate that glycosylation factor expression is tightly regulated and preserved across tissues and mammalian species, supporting the existence of a conserved glycosylation code.

### 3.3. Co-Expression Patterns Strengthen a Tissue-Specific Glycosylation Code

Since glycosylation patterns depend on the concerted activity of multiple glycosylation factors, one could expect some coordinated gene expression within tissues. Indeed, correlation analysis of human transcriptome profiles identified 6491 pairs of glycosylation factors highly correlated (Pearson R > 0.8) covering almost all glycosylation factors (Appendix A). Moreover, we identified glycosylation factors consistently co-expressed across several tissues (Figure 3A and Appendix A). The most common cluster includes a core of N-linked glycosylation genes *RPN1*, *RPN2*, *STT3A*, *GANAB* and *UGGT1*, which are strongly correlated to each other across over 10 tissues. Other cases include *FUT3* and *FUT6* in 10 tissues, and *OGT*, *OGA* and *ALG13* in nine tissues. Notably, only few glycosylation factors showed strong anti-correlated expressions (Pearson R < −0.8), such as ST3GAL3/*GALNT3* and ST3GAL3/*FUT3*, both in the colon and the esophagus (Appendix A). The consistent co-expression of glycosylation factors suggests that these are responsible for the biosynthesis of prevalent glycoconjugates common to several tissues.

Despite the conserved co-expression patterns, our analysis revealed an association between certain glycosylation subclasses and tissues. Indeed, we could depict several N-linked glycosylation genes highly correlated in muscle, and O-linked glycosylation genes in digestive system tissues (Appendix A). Such findings suggest a differential prevalence of glycosylation pathways in distinct tissues. More importantly, we identified two highly anti-correlated (R < −0.8) clusters of genes in the esophagus, indicating two opposing glycosylation patterns occurring in that tissue (Figure 3B). These opposing clusters included pairs of paralogous genes (e.g., *LARGE1* and *LARGE2*, *GALNT1* and *GALNT2*, *B3GNT8* and *B3GNT9*), indicating a concerted regulation of paralogous genes in the glycosylation process. Therefore, our findings unveiled the existence of a concerted expression of the glycosylation factors within tissues, reinforcing the existence of a firmly regulated glycosylation code.

### 3.4. Maintenance of the Glycosylation Machinery Is Important for Human Health

Due to such highly conserved tissue-specificity, one could foresee that impairment of glycosylation factors would have a significant impact on tissue homeostasis and human health. In fact, according to ClinVar resource [46], over 40% of the glycosylation factors enclose genetic variants associated with clinically pathogenic phenotypes (Figure 4A and Appendix A), overtaking the other classes of gene expression regulators. However, such proportion differs across the different glycosylation pathways, with pathogenic variants being described for 80% of the glypiation genes but only affecting 23% of the O-linked glycosylation factors (Figure 4B). The pathogenic variants in glycosylation factors are mostly associated with muscle and intellectual disability, usually included in the rare family of congenital disorders of glycosylation (CDGs) (Appendix A). The majority of CDGs are autosomal recessive disorders that manifest from infancy, being homozygous mutations usually associated with lethality to the embryo [47,48,49,50].

Due to the higher number of glycosylation factors associated with pathogenic variants, we decided to assess their intolerance to mutations in human samples using the pLI score (probability of being Loss-of-function intolerant). The pLI measures the probability of intolerance of a given gene to the loss of function on the basis of the frequency of protein-truncating variants [51]. Surprisingly, most of the glycosylation factors showed low or intermediated pLI scores, contrasting with other gene regulator classes (Appendix A). This apparent tolerance to protein variation may be due to compensatory co-expression of redundant paralogous glycosylation factors observed in certain tissues. Nevertheless, some factors showed high intolerance to loss of function mutations as it is the case of *STT3A*, *STT3B* and *GANAB* genes, belonging to the co-expressed cluster of N-glycosylation (Appendix A). Such findings disclose the genetic diversity, complexity and susceptibility underlying the glycosylation machinery.

Besides the impact of genetic alterations, glycosylation also changes during ageing, representing both a predisposition to and a functional mechanism involved in disease pathology [52]. In fact, GTEX expression profiles unveiled a significantly higher number of glycosylation factors altered in ageing samples of adipose tissue, artery, brain, and whole blood (Figure 4C and Appendix A, Appendix A). Interestingly, the expression of glycosylation factors seems to predominantly decrease with age in the brain, whole blood, and uterus, while it predominantly increases in adipose tissue and arteries (with a predominance of O-linked glycosylation) (Figure 4C). Apart from changing according to tissue-type and age, a smaller set of glycosylation factors showed sex-specific expression profiles in muscle, skin, thyroid, and adipose tissue (Appendix A). Overall, these results suggest that the glycosylation machinery may play a role not only in congenital anomalies but also in ageing-related diseases.

### 3.5. Genome and Transcriptome Alterations of Glycosylation Factors Are Pervasive in Cancer

One of the diseases with increasing prevalence with age is cancer, where aberrant glycosylation plays an important role [17]. However, pathogenic variants in glycosylation factors rarely appear as explicitly associated with cancer in relevant resources such as Clinvar (Appendix A) or IntOGen (Appendix A), and they do not seem to be under strong selective pressure in cancer (Appendix A). Accordingly, the genomic profiles from The Cancer Genome Atlas cohorts (TCGA) unveil that only a median of 0.34% of cancer patients have mutations in glycosylation factors, versus 0.66% and 0.54% for epigenetic and phosphorylation factors, respectively (Appendix A). Within glycosylation factors, glypiation genes emerge as the least mutation prone, with a median of only 0.2% of cancer patients harboring mutations in these genes (Appendix A). However, around 5% of cancer patients have copy number alterations (CNA) in the glycosylation factors, similarly to other gene classes (Appendix A). Some glycosylation factors showed genomic alterations (SNVs and CNVs) in 15% of tumor samples (namely *GPAA1*, *PIGZ* and *B3GNT5*) with a predominance of amplifications (Figure 5A). Other genes are mostly affected by deletions, such as *B3GALT6*. *PIGV* and *MAN1C1.* Interestingly, three of the ten most altered glycosylation genes in cancer are glypiation-related genes (*GPAA1*, *PIGZ*, *PIGX*), which are predominantly affected by amplifications (Figure 5A). *GPAA1*, *PIGZ* and *PIGX* amplifications are associated with worse prognosis at a pan-cancer level (Appendix A). CNAs are the most prevalent type of mutation even in the genes with most mutations, such as *FUT9* and *UGGT2*. Importantly, 26% of the glycosylation factors show a significant association between being affected by large genomic alterations and overall patient survival (Appendix A). Thus, disruption of glycosylation factors by large genomic alterations is frequent across several cancer types, where some can be potential prognostic biomarkers of tumor progression.

Besides genetic alterations, cancer progression is also associated with large transcriptome changes. Hence, similarly to the other regulators of gene expression, glycosylation factors are also shown to be perturbed at the transcriptional level, particularly in colon, lung, kidney and breast cancers from TCGA (Appendix A). Notably, a gene set enrichment analysis shows that glycosylation factors are predominantly upregulated in liver, breast, and lung cancers (Appendix A). Indeed, several glycosylation factors, namely *ALG8*, *ALG3*, *COLGATLT1*, *B4GALT3* and *DPM2,* show increased expression levels in these and other cancers (Figure 5B). In contrast, a limited number of genes are predominantly downregulated throughout all cancer types, such as *ST6GALNAC6*, *ST6GALNAC3*, *GALNT16* and *MAN1C1*. Interestingly, such transcriptome disruption appears to be driven at least in part by increased gene copy numbers, as there is a small but significant correlation (R = 0.3, *p* = 4.9 × 10^−10^) between CNA prevalence and expression alterations for the same gene (Appendix A). Indeed, the genomic amplification of *ALG3*, *B3GNT5*, *GPAA1*, *ST6GALNAC2*, *EXT1* leads to overexpression in tumors, whereas deletions of *PIGV*, *B3GNT7*, *B3GAT1*, *FUCA1*, *MAN1C1* genes drives to downregulation. Thus, our analysis unveils that glycosylation factors are disrupted in cancer by large genomic and transcriptomic alterations that may drastically affect glycan make-up in tumor cells.

### 3.6. Glycosylation Machinery Is Associated with Changes in the Cellular Composition of Healthy Tissues and Tumor Microenvironment

Cancer-associated changes in glycosylation may have an impact in cell-to-cell communication, namely between tumor cells and the immune system. For example, *B3GNT3*-mediated glycosylation of the immune checkpoint PD-L1 decreases the capacity of the immune system to suppress tumor progression [53]. Consistently, our analysis unveiled that *B3GNT3* amplification in tumors was associated with the worst patient survival (Appendix A). Thus, to deeply characterize the immune-cell populations associated with glycosylation factor disruption in cancer, we explored cellular compositions inferred by RNA-based deconvolution methods [37]. Combining cellular compositions and gene expression levels within TCGA tumor samples, we depicted glycosylation factors consistently associated with an increase of specific immune-cell types across several cancer types (Figure 6A). The data showed that the presence of dendritic and natural killer cells was associated with the alterations mostly in O-linked glycosylation genes (e.g., *B4GAT1*, *POMT1*, *POMT2*, *RXYLT1*, *TMTC1*, *FUT9*, *GALNT11*, *GALNT13*). In opposition, the presence of macrophages, B and T-cells is favored by alterations in different glycosylation gene subclasses (e.g., *ST8SIA4*, *DSE*, *MFNG*, *FUT7*, *MAN1C1* and *ST8SIA1*). Interestingly, the expression of sialyltransferases, particularly of the ST8SIA family, is most often associated with changes in the frequencies of tumor-associated immune cell populations (Figure 6A). For example, in colorectal cancer, an increase in the expression of *ST8SIA4*, *ST3GAL6*, *ST6GALNAC5* is associated with an increase in regulatory T cells and pro-tumoral M2 macrophages (Figure 6B). In addition, a lasso regression analysis revealed that the expression of *ST8SIA4* and other glycosylation factors is positively correlated with pro-tumoral M2 macrophages in various tumor types (Appendix A). Thus, our pan-cancer analysis suggests that disruption of glycosylation factors may lead to specific immune modulation responses in the tumor microenvironment.

Given that glycosylation patterns may also influence the immune cell composition in healthy tissues, we also combined gene expression levels and cellular composition within GTEX tissues. The correlation analysis revealed several significant positive associations between glycosylation factors levels and immune cell populations in healthy tissues, particularly with pro-inflammatory M1 macrophages (Appendix A). Interestingly, we identified a positive association between the *DSE* expression and the relative frequency of M1 macrophages, both in normal (16 distinct GTEX tissues, Appendix A) and tumor tissues (five different TCGA tumor types, Figure 6A). In fact, the *DSE* gene codifies an glycosaminoglycan isomerase, acting as a tumor-rejection antigen and with the potential to stimulate anti-tumoral immunoreactivity [54].

Besides immune-system crosstalk, glycan structures also mediate cell-to-cell communication in the tumor microenvironment influencing tumor progression, where different cancer cell subclones evolve and co-exist (designated as intra-tumor heterogeneity). Thus, we combined the TCGA genomic profiles to infer the number of genetically distinct subclones within each sample (see Materials and Methods). Our lasso regression analysis depicted genomic and transcriptome alterations of several glycosylation factors associated with intra-tumor heterogeneity levels across several cancer types (Figure 6C). Interestingly, some molecular alterations showed the same recurrent outcome in several cancer types. Namely, increased *B3GNT4* expression was associated with subclonal expansion in at least seven tumor types, while *GALNT16* and *GALNT17* levels appeared mostly associated with a decrease in intra-tumor heterogeneity across five cancers. In summary, our analysis revealed that specific alterations in the glycosylation machinery are associated with the presence of distinct immune-cell populations in tumor/normal tissues, and also with tumor subclone diversity.

## 4. Discussion

Here, we used computational approaches and genome/transcriptome profiling to deeply characterize the evolution and relevance of the glycosylation machinery in human health. Our study unveiled that glycosylation factors are an ancient and conserved family of genes, with an apparent evolutionary pressure to keep a low number of copies, particularly in GPI and N-linked glycosylation, with most glypiation genes being single copies. Despite the low copy number, glycosylation factors also showed high tolerance of loss-of-function mutations, suggesting that a single functional allele suffices for their function or there is a functional compensation from paralogous genes. This is consistent with the etiology of the congenital disorders of glycosylation (CDG), where most tend to be autosomal recessive and enclose less than 100 cases [55]. Accordingly, CDG animal models tend to be embryonic lethal, with the rare instances of viability resulting from hypomorphic variants [56,57] Noteworthy, most genes associated with known CDG are ubiquitously expressed in healthy human tissues, in line with the observation that most CDGs display complex pleiotropic clinical phenotypes [47]. O-linked and lipid glycosylation are exceptions to this evolutionary path, with a few gene families enclosing large numbers of copies. Perhaps unsurprisingly, these genes display the greatest degree of tissue-specificity expression, likely contributing to the formation of distinct glycosylation patterns in tissues.

Our results support the existence of a tissue-specific glycosylation code established by the coordinated expression of distinct glycosylation factors. Indeed, the gene expression pattern of glycosylation factors enables the correct classification of different tissues, almost similar to that of transcription and phosphorylation factors; already known to be essential for tissue identity [58]. Within glycosylation factors, O-linked and lipid glycosylation showed the highest ability to distinguish different tissues, suggesting an important role for such specific post-translational modifications in tissue definition. Moreover, we unveiled that such specific patterns are phylogenetically conserved across mammals’ tissues, reinforcing the existence of tissue-specific glycosylation code. Nevertheless, we acknowledge that the transcriptome may only partially correlate with the activity of glycosylation factors and effective changes in glycan profiles [59]. Despite vast progress in the field of glycoproteomics, these technologies still do not afford a very detailed topology of the glycostructures [1], making the link with gene expression even more complicated. To our knowledge, there is no map of glycostructures simultaneously generated for several human tissues that would enable us to systematically compare with expression data from protein atlas or GTEX at a larger scale, and even less quantitatively. A study in zebrafish has shown that the presence of tissue-specific glycosylation patterns such as sialylation could be associated with gene expression of enzymes associated with sialylation [60]. A recent study in the mammalian brain also observes a correlation between abundances of some glycostructures and gene expression [61]. Finally, recent studies have simultaneously assess the presence of glycan structures and transcripts at the single-cell level, combining scRNA-Seq with one [62] or several [63] lectin-bound DNA-barcodes covering different type of glycosylated proteins. These technologies enabled establishing correlation between gene expression and glycan structures as it is case of ST6Gal1 which showed the highest correlation with α2-6Sia-binding lectin rPSL1a [63]. Yet, the accessibility and resolution of these methods is still quite low. Regarding the different tissues, brain showed the highest number of specific glycosylation factors, in agreement with neuromuscular manifestations being the most frequently described clinical phenotypes derived from glycosylation defects [47]. Recent studies show a very distinctive restricted glycosylation repertoire in the brain, suggesting tight regulation [61,64]. In our analysis, we observed several brain-specific GALNTs (GALNT13/15/17), in agreement with the observation of the predominance of O-GalNAc structures in the brain [61]. Furthermore, we depicted tissue-specific clusters of strongly correlated O-linked and lipid glycosylation factors in stomach, intestine, and colon, providing further evidence of the potential importance of these pathways in the digestive system. Indeed, O-glycosylation of intestinal mucins are among the most well studied examples of the biological relevance of glycosylation, where mucins play a fundamental role in host-microbiota interactions [65,66]. Therefore, we unveiled tissue-specific clusters of co-expressed glycosylation factors, suggesting a synergistic and/or independent role of different glycosylation pathways in the various tissues. More strikingly, we could also detect in the esophagus two strongly antagonistic clusters of glycosylation factors (mostly associated with O-linked and lipid glycosylation), both sharing a common, mostly independent, core of N-linked glycosylation. One of these clusters contain the *LARGE1*/*B4GAT1* unit responsible for the glycosylation of dystroglycan [44], and several elements of the canonical and non-canonical O-linked glycosylation pathway, including *COLGALT1*, *POMT2*, *POFUT2*, and *POGLUT2* [67]. The other cluster contains several elements of the Golgi-associated O-linked glycosylation of mucins, including several B3GNTs, GALNTs, and *C1GALT1*.

The existence of a tissue-specific glycosylation code reinforces the importance of this post-translational modification process in tissue homeostasis throughout all developmental stages, including in adulthood. Ageing is among the factors most associated with increased cellular dysfunction and emergence of morbidities [68]. In fact, alterations in immunoglobin G glycosylation have been associated with age-related diseases such as diabetes and hypertension [52]. Interestingly, a commercial test (GlycoAgeTest©) uses the logarithm of the ratio of two N-glycans (NGA2F and NA2F) to infer the glycosylation age of a person [69,70], similarly to the methylation clock [71]. Our exploration of transcriptome profiles of GTEX project unveiled a significant down-regulation of the sialic acid transporter SLC35A1 in blood samples from older people, in agreement with the reported decrease of sialylation with age [68]. This profile also showed down-regulation of the galactosyltransferase B3GALT4 and the sialyltransferase ST3GAL2, suggesting a decrease in galactosylation and sialylation in O-glycan and GSL with age, yet to be identified. Therefore, our results suggest a downregulation of the glycan structures, derived from *B3GALT4*, *ST3GAL2* and/or *SLC35A1* genes, with ageing, a feature so far only described in neurodegenerative diseases [72]. Moreover, we detected a significant increase in the expression of the mannosidase *MAN1C1* gene in the ageing-skin, an alteration that has also been reported in a previous study [73]. Interestingly, we also observed an increased expression of several O-linked glycosylation factors in the arteries and adipose tissue of old people, suggesting a role of this pathway in ageing-related alterations in these tissues. Previous works have also shown that ageing is associated with a decreased expression of several glycosylation factors in the brain [74]. Since most studies assessing ageing-related changes in glycosylation use blood derivatives (plasma, serum, immunoglobulin fractions), their applicability may be limited and overlook tissue-specific effects. Thus far, our findings further indicate that age-associated changes in glycosylation are tissue-specific.

Alterations in the glycosylation machinery have been previously detected in cancer [17]. Our pan-cancer analysis of TCGA cohorts supports the idea that short variants directly affecting gene function are not likely to be a major mechanism underlying defects in glycosylation leading to cancer [67,75]. Instead, they suggest that copy number alterations of glycosylation factors are quite pervasive across all cancer types, with a predominance of genomic amplifications. These findings are consistent with previous global analysis showing that large genomic alterations are more recurrent in cancer than small genetic alterations [76]. In addition, our analysis unveiled that such genomic alterations are reflected at the transcriptome level and may affect the glycan make-up in tumor cells. More importantly, we can depict several genomic alterations of specific glycosylation factors associated with changes in patient survival, which can be potential biomarkers for prognosis. In agreement, glycosyltransferases have been shown to be useful in the classification and prognosis of certain cancers [77], where differences in the expression of sialyltransferases seem to be among the most relevant to distinguish different cancer types. An increase in sialylation has been reported in some tumor types, such as colorectal cancer [17,74].

Alteration in the glycosylation patterns of tumor cells plays a very important in the modulation of immune responses to cancer and have been shown to be major factors in the design of novel immunotherapies [78]. When we combined cancer transcriptome profiles and estimates of immune-cell abundance (through RNA-based deconvolution approaches), we observed a recurrent association between the expression levels of specific glycosylation subclasses and the presence of dendritic, natural killer cells, macrophages, B and T-cells. In fact, we detected an increase in the expression of sialylation genes and the presence of regulatory T-cells and pro-tumoral M2 macrophages, in agreement with recent reports linking sialylation to poorer cancer outcomes [18,79,80]. We could also detect a recurrent association between changes in the gene expression of some glycosylation factors and intra-tumoral heterogeneity, the relevance of which is yet to be determined.

In conclusion, through exploration of molecular profiles from healthy, ageing and diseased tissues, we showed the potential of glycosylation factors in defining tissue identity through a tissue-specific glycosylation code, analogous to transcription and phosphorylation. Moreover, our work unveiled that significant alterations in the gene expression patterns of the glycosylation machinery occur in ageing-tissues, suggesting that cell-specific glycoconjugate profiles may change throughout adulthood during the aging process. Finally, large copy number amplifications and corresponding changes in the expression of glycosylation factors emerged as the main associators to cancer-related alterations.

Overall, this work provides a rich set of information, derived from multiple and integrative sources, that will be useful for the glycosciences community in expanding the knowledge on the function of glycosylation in human health.

## Figures and Tables

**Figure 2 biomedicines-10-01805-f002:**
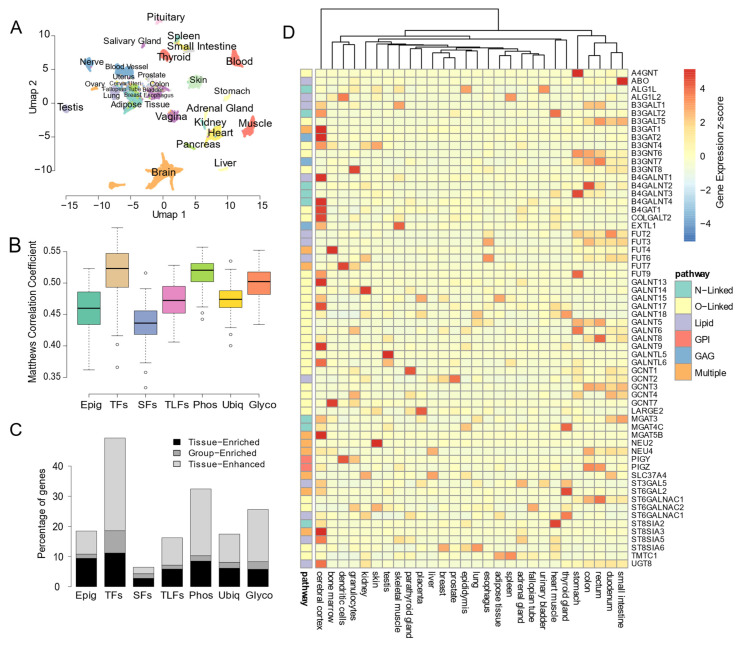
Glycosylation Factors show tissue-specific expression. (**A**) Uniform Manifold Approximation and Projection (UMAP) representation, based on the gene expression profiles of glycosylation factors of GTEX samples, colored according to their tissue of origin. (**B**) Boxplot representation of the Matthews Correlation Coefficient (MCC) value for the classification of the tissue of each GTEX sample (*n* = 100 random forest models using 30 randomly selected genes of each gene class). (**C**) Cumulative bar plot representation of the frequency of genes belonging to each gene class that show tissue-specific expression in protein atlas (colors represent degree of specificity as defined in TissueEnrich from the most specific Tissue-Enriched to the least specific Tissue-Enhanced). (**D**) Heatmap representation of the protein atlas gene expression of glycosylation factors displaying some degree of specificity (represented expression values are scaled z-scores). Only tissues with at least one gene with z-score > 2 are represented.

**Figure 3 biomedicines-10-01805-f003:**
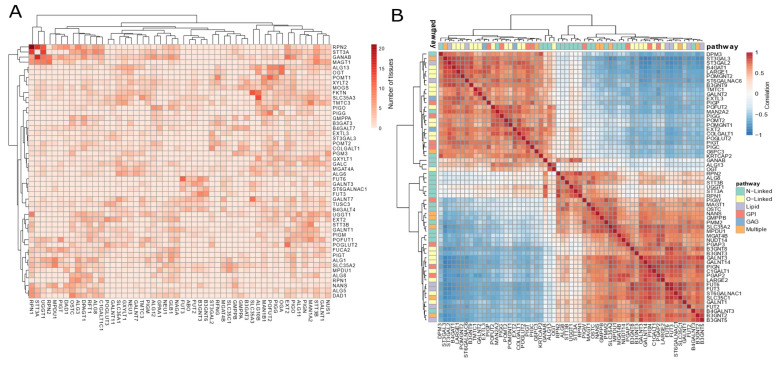
Glycosylation factors show co-expression clusters across healthy tissues. (**A**) Heatmap representation of the number of tissues with high positive correlation between pairs of glycosylation factors (based on the gene expression of GTEX samples, R > 0.8, *p* < 0.05). (**B**) Heatmap representation of the correlation values of highly (anti)correlated pairs of glycosylation factors (based on the gene expression of GTEX esophagus samples, |R| > 0.8. Node colors represent glycosylation subclass; color intensity and line thickness represent the number of tissues where the gene pair displays a high degree of correlation.

**Figure 4 biomedicines-10-01805-f004:**
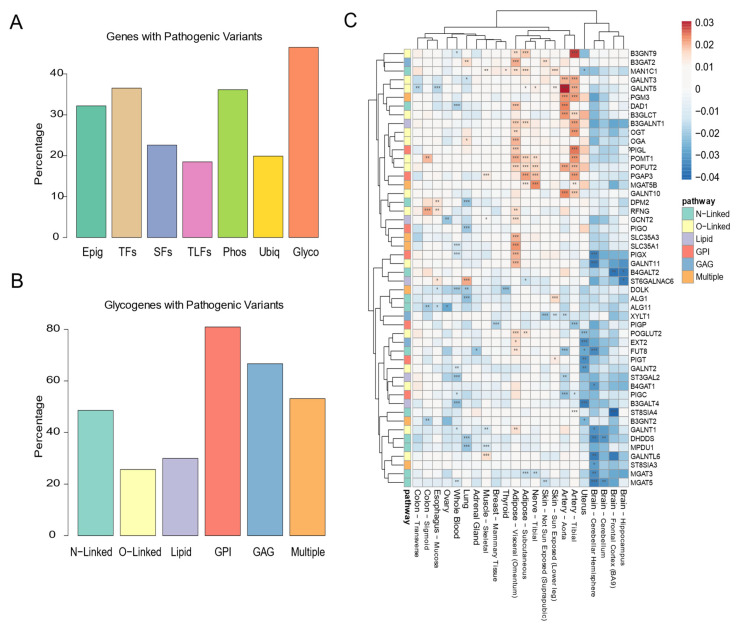
Glycosylation machinery is important in human health. (**A**) Bar plot of the percentage of genes associated with pathogenic variants in Clinvar, according to their gene class. (**B**) Bar plot of the percentage of glycosylation factors associated with pathogenic variants in Clinvar, according to their glycosylation subclass. (**C**) Heatmap representation of glycosylation factors whose expression changes in GTEX ageing tissues. Heatmap colors represent the coefficients from a linear model associating age with gene expression, in a given tissue (*: *p* < 0.05; **: *p* < 0.01; ***: *p* < 0.001). Red indicates higher and blue lower expression with ageing.

**Figure 5 biomedicines-10-01805-f005:**
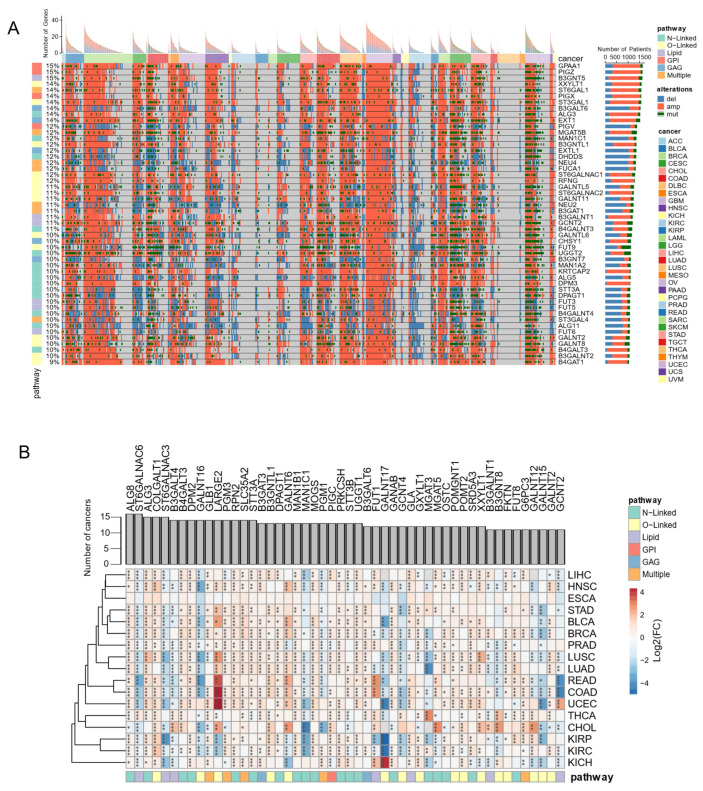
Glycosylation Factors are dysregulated in cancer. (**A**) Oncoprint view of the 50 glycosylation factors with highest frequency of genetic alterations (over all TCGA samples). (**B**) Heatmap representation of the transcriptome alterations (log2-fold change) between tumor and normal TCGA samples, of the 40 glycosylation factors most frequently deregulated in the different tumor types (*: *p* < 0.05; **: *p* < 0.01; ***: *p* < 0.001). ACC (adrenocortical carcinoma); BLCA (bladder urothelial carcinoma); BRCA (breast invasive carcinoma); CESC (cervical squamous cell carcinoma and endocervical adenocarcinoma); CHOL (cholangiocarcinoma); COAD (colon adenocarcinoma); DLBC (Lymphoid Neoplasm Diffuse Large B-cell Lymphoma); ESCA (Esophageal carcinoma); GBM (Glioblastoma multiforme); HNSC (head and neck squamous cell carcinoma); KICH (kidney Chromophobe); KIRC (kidney renal clear cell carcinoma), KIRP (kidney renal papillary cell carcinoma); LAML (Acute Myeloid Leukemia); LGG (Brain Lower-Grade Glioma); LIHC (liver hepatocellular carcinoma); LUAD (lung adenocarcinoma); LUSC (lung squamous cell carcinoma), MESO (Mesothelioma); OV (Ovarian serous cystadenocarcinoma); PAAD (pancreatic adenocarcinoma); PCPG (pheochromocytoma and Paraganglioma); PRAD (prostate adenocarcinoma); READ (rectum adenocarcinoma); SARC (Sarcoma); SKCM (Skin Cutaneous Melanoma); STAD (Stomach adenocarcinoma); TGCT (Testicular Germ Cell Tumors); THCA (thyroid carcinoma); THYM (Thymoma); UCEC (uterine Corpus Endometrial Carcinoma); UCS (Uterine Carcinosarcoma); and UVM (Uveal Melanoma).

**Figure 6 biomedicines-10-01805-f006:**
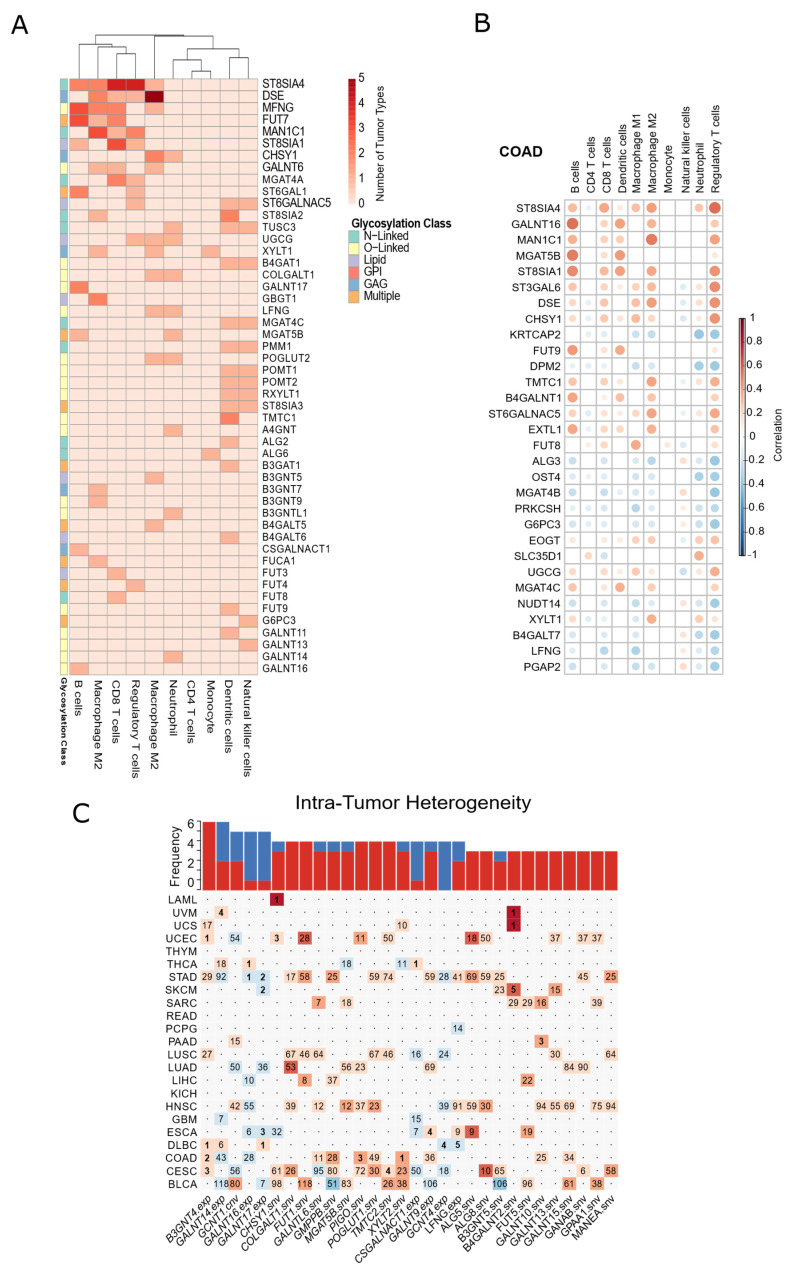
Glycosylation Factors have prognostic value in cancer. (**A**) Heatmap representation of the number of TCGA cancer types where there is a significant correlation (FDR < 0.05) between gene expression and frequencies of immune cell populations (obtained from TCIA). (**B**) Heatmap representation of Pearson correlation values between gene expression values of glycosylation factors and estimated frequency of immune cell populations (from TCIA), in TCGA COAD samples (only significant—*p* < 0.05—correlation values are displayed, and only top 30 genes with highest (anti)correlation are displayed). (**C**) Heatmap representation of the significant coefficients for gene perturbation events in a lasso-regression model correlating events with the number of genetic clones (estimated using EXPANDS [41]) in TCGA samples. Red color indicates positive coefficients and blue color, negative coefficients. Color intensity indicates value of the coefficient, with darker tones indicating higher values for the coefficient. Numbers indicate relative importance of the event in the regression model. Only cancers with an overall R > 0.3 for the model are displayed.

## Data Availability

All data used in this work are publicly accessible and referred to in the methods or text as appropriate.

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
