# Peer review of "Concerted Regulation of Glycosylation Factors Sustains Tissue Identity and Function"

_biomedicines, 2022, doi:10.3390/biomedicines10081805_

Round 1
Reviewer 1 Report
The manuscript of Sobral et. al. thoroughly explores the information publicly available on the glycan genomes and transcriptomes in various tissues, and across age, sex and diseases. This is important because glycogenes are often included in in large omics dataset, but they do usually not get the attention deserved. The manuscript is well written and the figures supplement the text. However, it was hard to evaluate the main manuscript figures, as the resolution was in most cases too low to properly read the text on the axis of the plots. This should be solved in the final format.
While the work is extensive and makes good use of the available omics resources, I do have some concerns before considering the work ready for publication. Especially the discussion requires some critical evaluation of some of the statements made:
1) Throughout the text, starting with the title, the authors overstate the causal effect between altered glycosylation factors and effects on health. The current study only describes associations between glycogenes (expression) and health factors, but does not investigate causal relationships. As there are many steps between genetic or transcriptomic alterations and final glycan appearance (protein or lipid carrier biosynthesis, glycan biosynthesis, glycocunjugate localization, to name few), and because the changes in glycosylation factors can also be an effect of a disease rather than a cause, one cannot make statements on the causal effect of the alterations observed in this study. Examples that should be reevaluated:
a. Title: “influences human health”
b. Line 569: “In fact, pro-inflammatory glycosylation patterns in the blood (e.g. in IgG) increase in ageing, which may explain some age-related diseases such as diabetes and hypertension.”
c. Line 589: “Alterations in the glycosylation machinery during adulthood may prompt the appearance of ageing-related diseases, such as cancer.”
d. Line 620: “Moreover, our work unveiled that the impairment of the glycosylation machinery occurs in ageing-tissues, suggesting a role in ageing-related.” (This sentence also seems unfinished)
e. Line 622: “Finally, large copy number amplifications and corresponding changes in the expression of glycosylation factors emerged as the main players of glycosylation-driven tumorigenesis.”
2) The relationship between genetic and transcriptomic changes and the actual effect on glycan presentation is not discussed thoroughly enough. Although the authors state: “Nevertheless, we acknowledge that the transcriptome may only partially correlate with the activity of glycosylation factors and effective changes in glycan profiles.” (line 544), this is an important point and can be expanded based on literature.
a. There are several recent reports investigating the relationship between glycogene expression and glycan presentation in a cellular system. These can be included in the discussion.
b. Many N- and O-glycomics and glycoproteomics studies are available, describing glycan representation in different tissues. When discussing the altered expression between tissues, e.g. in line 546: “Regarding the different tissues, brain showed the highest number of specific glycosylation factors, in agreement with neuromuscular manifestations being the most frequently described clinical phenotypes derived from glycosylation defects.”, this should be linked to literature describing glycomics and glycoproteomics of these type of tissues. Are similar drastic differences observed?
3) Line 573: “Our exploration of transcriptome profiles of GTEX project unveiled a significant down-regulation of the galactosyltransferase B3GALT4, the sialyltransferase ST3GAL2 and the sialic acid transporter SLC35A1 in blood samples from older people, in agreement with the reported decrease of galactosylation and sialylation with age[68].” As the sialyltransferase ST3GAL2 has been reported to be responsible for O-glycan and GSL sialylation (and not N-linked glycan sialylation) I don’t think the link between this enzyme and aging related sialylation changes can be made. At least it is not supported by the indicated reference 68. Likewise, galactosyltransferase B3GALT4 is part of the GSL-pathway, and should not be related to N-glycan galactosylation changes with age.
4) Line 579: “Moreover, we detected a significant increase in the expression of the mannosidase MAN1C1 in ageing-skin, where alterations in mannosylation have been previously reported[70].” Reference 70 does not support this claim.
Minor comments:
1) Line 20: “These were categorized as involved in N-linked, and O-linked glycosylation, and glypiation, lipid glycosylation, and glycosaminoglycan.” Suggestion: These were categorized as involved in N-, O- and lipid-linked glycosylation, glypiation, and glycosaminoglycan synthesis.
2) Two .. at the end of the abstract
3) Line 81: the word glycosylation is missing in this sentence.
4) The study describes to do multi-omics, which might be considered a slight overstatement as only two omics are included.
Author Response
We thank the Referees for their thorough review of our manuscript and their constructive
comments and suggestions. We present herein our point-by-point rebuttal for your evaluation
and the revised manuscript.
Point-by Point response to Reviewer #1
The manuscript of Sobral et. al. thoroughly explores the information publicly available on the
glycan genomes and transcriptomes in various tissues, and across age, sex and diseases.
This is important because glycogenes are often included in in large omics dataset, but they do
usually not get the attention deserved. The manuscript is well written and the figures
supplement the text. However, it was hard to evaluate the main manuscript figures, as the
resolution was in most cases too low to properly read the text on the axis of the plots. This
should be solved in the final format.
Response: We would like to thank the reviewer for her/his careful evaluation of our manuscript
and appreciate the overall positive comments. We have improved the resolution of all main
manuscript figures.
While the work is extensive and makes good use of the available omics resources, I do have
some concerns before considering the work ready for publication. Especially the discussion
requires some critical evaluation of some of the statements made:
1) Throughout the text, starting with the title, the authors overstate the causal effect between
altered glycosylation factors and effects on health. The current study only describes
associations between glycogenes (expression) and health factors, but does not investigate
causal relationships. As there are many steps between genetic or transcriptomic alterations
and final glycan appearance (protein or lipid carrier biosynthesis, glycan biosynthesis,
glycocunjugate localization, to name few), and because the changes in glycosylation factors
can also be an effect of a disease rather than a cause, one cannot make statements on the
causal effect of the alterations observed in this study. Examples that should be reevaluated:
Response: We understand the reviewer’s comments and have rewritten the examples
mentioned bellow, including the evaluations made at the discussion.
a. Title: “influences human health”
Response: We have rewritten the title accordingly:
“Concerted regulation of glycosylation factors sustains tissue identity and function”
b. Line 569: “In fact, pro-inflammatory glycosylation patterns in the blood (e.g. in IgG)
increase in ageing, which may explain some age-related diseases such as diabetes and
hypertension.”
Response: We have rewritten the sentence accordingly:
“In fact, alterations in immunoglobin G glycosylation have been associated with age-related
diseases such as diabetes and hypertension.”
c. Line 589: “Alterations in the glycosylation machinery during adulthood may prompt the
appearance of ageing-related diseases, such as cancer.”
Response: We have rewritten the sentence accordingly:
“Alterations in the glycosylation machinery have been previously detected in cancer [17].”
d. Line 620: “Moreover, our work unveiled that the impairment of the glycosylation
machinery occurs in ageing-tissues, suggesting a role in ageing-related.” (This sentence also
seems unfinished)
Response: We have rewritten the sentence accordingly:
“Moreover, our work unveiled that significant alterations in the gene expression patterns of the
glycosylation machinery occurs in ageing-tissues, suggesting that cell-specific glycoconjugate
profiles may change throughout adulthood during the aging process.”
e. Line 622: “Finally, large copy number amplifications and corresponding changes in the
expression of glycosylation factors emerged as the main players of glycosylation-driven
tumorigenesis.”
Response: We have rewritten the sentence accordingly:
“Finally, large copy number amplifications and corresponding changes in the expression of
glycosylation factors emerged as the main associators to cancer-related alterations”
2) The relationship between genetic and transcriptomic changes and the actual effect on
glycan presentation is not discussed thoroughly enough. Although the authors state:
“Nevertheless, we acknowledge that the transcriptome may only partially correlate with the
activity of glycosylation factors and effective changes in glycan profiles.” (line 544), this is an
important point and can be expanded based on literature.
a. There are several recent reports investigating the relationship between glycogene
expression and glycan presentation in a cellular system. These can be included in the
discussion.
Response: We acknowledge and agree with the reviewer’s concern. We have extended the
discussion to incorporate some further attention to this issue.
“Despite vast progresses in the field of glycoproteomics, these technologies still do not afford
a very detailed topology of the glycostructures [1], making the link with gene expression even
more complicated. To our knowledge, there is no map of glycostructures simultaneously
generated for several human tissues that would enable us to systematically compare with
expression data from protein atlas or GTEX at a larger scale, and even less quantitatively. A
study in zebrafish has shown that the presence of tissue-specific glycosylation patterns such
as sialylation could be associated with gene expression of enzymes associated with sialylation
[60]. A recent study in the mammalian brain also observes a correlation between abundances
of some glycostructures and gene expression [61]. Finally, recent studies have
simultaneously assess the presence of glycan structures and transcripts at the single-cell
level, combining scRNA-Seq with one [62] or several [63] lectin-bound DNA-barcodes
covering different type of glycosylated proteins. These technologies enabled establishing
correlation between gene expression and glycan structures as it is case of ST6Gal1 which
showed the highest correlation with α2-6Sia-binding lectin rPSL1a [63]. Yet, the accessibility
and resolution of these methods is still quite low.”
b. Many N- and O-glycomics and glycoproteomics studies are available, describing glycan
representation in different tissues. When discussing the altered expression between tissues,
e.g. in line 546: “Regarding the different tissues, brain showed the highest number of specific
glycosylation factors, in agreement with neuromuscular manifestations being the most
frequently described clinical phenotypes derived from glycosylation defects.”, this should be
linked to literature describing glycomics and glycoproteomics of these type of tissues. Are
similar drastic differences observed?
Response: We thank the reviewer for the constructive opinion. We’ve included some literature
focusing on glycosylation in the brain.
“Recent studies show a very distinctive restricted glycosylation repertoire in the brain,
suggesting a tight regulation [61,64]. In our analysis, we observed several brain-specific
GALNTs (GALNT13/15/17), in agreement with the observation of the predominance of OGalNAc structures in the brain [61].”
3) Line 573: “Our exploration of transcriptome profiles of GTEX project unveiled a significant
down-regulation of the galactosyltransferase B3GALT4, the sialyltransferase ST3GAL2 and
the sialic acid transporter SLC35A1 in blood samples from older people, in agreement with
the reported decrease of galactosylation and sialylation with age[68].” As the sialyltransferase
ST3GAL2 has been reported to be responsible for O-glycan and GSL sialylation (and not Nlinked glycan sialylation) I don’t think the link between this enzyme and aging related sialylation
changes can be made. At least it is not supported by the indicated reference 68. Likewise,
galactosyltransferase B3GALT4 is part of the GSL-pathway, and should not be related to Nglycan galactosylation changes with age.
Response: Thank you for raising this question. we agree with the reviewer and have corrected
the sentence to the following:
“Our exploration of transcriptome profiles of GTEX project unveiled a significant downregulation of the sialic acid transporter SLC35A1 in blood samples from older people, in
agreement with the reported decrease of sialylation with age [68]. This profile also showed
down-regulation of the galactosyltransferase B3GALT4 and the sialyltransferase ST3GAL2
suggesting a decrease of galactosylation and sialylation in O-glycan and GSL with age, yet to
be identified."
4) Line 579: “Moreover, we detected a significant increase in the expression of the
mannosidase MAN1C1 in ageing-skin, where alterations in mannosylation have been
previously reported[70].” Reference 70 does not support this claim.
Response: We thank the reviewer for calling our attention to this issue. We have corrected
the sentence to the following:
“Moreover, we detected a significant increase in the expression of the mannosidase MAN1C1
gene in the ageing-skin, an alteration that has also been reported in a previous study [74].”
Minor comments:
1) Line 20: “These were categorized as involved in N-linked, and O-linked glycosylation,
and glypiation, lipid glycosylation, and glycosaminoglycan.” Suggestion: These were
categorized as involved in N-, O- and lipid-linked glycosylation, glypiation, and
glycosaminoglycan synthesis.
Response: We corrected the sentence as suggested.
2) Two .. at the end of the abstract
Response: We corrected the dots as suggested.
3) Line 81: the word glycosylation is missing in this sentence.
Response: We add the word glycosylation in the sentence.
4) The study describes to do multi-omics, which might be considered a slight overstatement
as only two omics are included.
Response: We understand the reviewer’s comments and replaced the expression multi-omics
by genome/transcriptome.
Reviewer 2 Report
This manuscript talked about the glycosylation related genes collected through GGDB glycosylation factor database, Reactome and KEGG database, then analyzed their tissue-specific expression in healthy human tissues, and the expression in all kinds of cancers, indicating that glycosylation is tightly regulated in healthy tissues but impaired in glycosylation. This work is interesting and meaning for, but there are many points should be further considered.
1. The glycosylation-factors and gene expression related genes are selected through GGDB glycosylation factor database, Reactome and KEGG database. What is the basis for screening related genes? How to evaluate their reliability?
2. As we know, glycosylation is a post-translation modification of protein, but the manuscript did not consider the changes in protein level, just analyzed the gene expression of glycosylation related genes in healthy tissue and cancers. I wonder whether the conclusion is believable and persuasive based solely on gene expression when we talked about a post-translation modification of protein.
Author Response
We thank the Referees for their thorough review of our manuscript and their constructive
comments and suggestions. We present herein our point-by-point rebuttal for your evaluation
and the revised manuscript.
Point-by Point response to Reviewer #2
This manuscript talked about the glycosylation related genes collected through GGDB
glycosylation factor database, Reactome and KEGG database, then analyzed their tissuespecific expression in healthy human tissues, and the expression in all kinds of cancers,
indicating that glycosylation is tightly regulated in healthy tissues but impaired in glycosylation.
This work is interesting and meaning for, but there are many points should be further
considered.
Response: We would like to thank the reviewer for her/his careful evaluation of our manuscript
and appreciate the overall positive comments.
1. The glycosylation-factors and gene expression related genes are selected through GGDB
glycosylation factor database, Reactome and KEGG database. What is the basis for screening
related genes? How to evaluate their reliability?
Response: We thank the reviewer for the question. We screened related genes because we
wanted to compare glycosylation factors, as a group, with other groups of genes also known
to be important in defining cellular identity. This includes transcription factors and epigenetic
factors, for which there were high quality curated lists of genes in the literature that we could
use. Then, we also included other groups of genes important for gene expression regulation,
namely translation, splicing, phosphorylation, and ubiquitination. Particularly in the later
groups, where we used information from gene ontology annotations, we may have missed
some of their elements, but to classify genes in those groups we used highly specific
annotation terms, and therefore the genes that were included are highly likely to be good
representatives for the comparison. In the manuscript, we have explained that the goal was to
understand whether glycosylation factors or gene expression related genes distinguish the
different tissues.
2. As we know, glycosylation is a post-translation modification of protein, but the manuscript
did not consider the changes in protein level, just analyzed the gene expression of
glycosylation related genes in healthy tissue and cancers. I wonder whether the conclusion is
believable and persuasive based solely on gene expression when we talked about a posttranslation modification of protein.
Response: We acknowledge and agree with the reviewer’s concern. We have tried to bring
more attention to this issue by extending the discussion, by including the following paragraph:
“Nevertheless, we acknowledge that the transcriptome may only partially correlate with the
activity of glycosylation factors and effective changes in glycan profiles[59]. Despite vast
progresses in the field of glycoproteomics, these technologies still do not afford a very detailed
topology of the glycostructures [1], making the link with gene expression even more
complicated. To our knowledge, there is no map of glycostructures simultaneously generated
for several human tissues that would enable us to systematically compare with expression
data from protein atlas or GTEX at a larger scale, and even less quantitatively. A study in
zebrafish has shown that the presence of tissue-specific glycosylation patterns such as
sialylation could be associated with gene expression of enzymes associated with sialylation
[60]. A recent study in the mammalian brain also observes a correlation between abundances
of some glycostructures and gene expression [61]. Finally, recent studies have
simultaneously assess the presence of glycan structures and transcripts at the single-cell
level, combining scRNA-Seq with one [62] or several [63] lectin-bound DNA-barcodes
covering different type of glycosylated proteins. These technologies enabled establishing
correlation between gene expression and glycan structures as it is case of ST6Gal1 which
showed the highest correlation with α2-6Sia-binding lectin rPSL1a [63]. Yet, the accessibility
and resolution of these methods is still quite low.”